# Cellular and Molecular Mechanisms and Effects of Berberine on Obesity-Induced Inflammation

**DOI:** 10.3390/biomedicines10071739

**Published:** 2022-07-19

**Authors:** Ji-Won Noh, Min-Soo Jun, Hee-Kwon Yang, Byung-Cheol Lee

**Affiliations:** Department of Clinical Korean Medicine, Graduate School, Kyung Hee University, 26 Kyungheedae-ro, Dongdaemun-gu, Seoul 02447, Korea; oiwon1002@khu.ac.kr (J.-W.N.); jms3798@khu.ac.kr (M.-S.J.); yang2019@khu.ac.kr (H.-K.Y.)

**Keywords:** berberine, obesity, inflammation, adipose tissue macrophage, chemotaxis

## Abstract

Obesity represents chronic low-grade inflammation that precipitates type 2 diabetes, cardiovascular disease, and cancer. Berberine (BBR) has been reported to exert anti-obesity and anti-inflammatory benefits. We aimed to demonstrate the underlying immune-modulating mechanisms of anti-obesity effects of BBR. First, we performed in silico study to identify therapeutic targets, describe potential pathways, and simulate BBR docking at M1 and M2 adipose tissue macrophages (ATMs), tumor necrosis factor-α (TNF-α), C-C motif chemokine ligand 2 (CCL2), CCL4, CCL5, and C-X-C motif chemokine receptor 4 (CXCR4). Next, in vivo, we divided 20 C58BL/6 mice into four groups: normal chow, control (high fat diet (HFD)), HFD + BBR 100 mg/kg, and HFD + metformin (MET) 200 mg/kg. We evaluated body weight, organ weight, fat area in tissues, oral glucose and fat tolerance tests, HOMA-IR, serum lipids levels, population changes in ATMs, M1 and M2 subsets, and gene expression of TNF-α, CCL2, CCL3, CCL5, and CXCR4. BBR significantly reduced body weight, adipocyte size, fat deposition in the liver, HOMA-IR, triglycerides, free fatty acids, ATM infiltration, all assessed gene expression, and enhanced the CD206+ M2 ATMs population. In conclusion, BBR treats obesity and its associated metabolic dysfunctions, by modulating ATM recruitment and polarization via chemotaxis inhibition.

## 1. Introduction

Obesity is part of the metabolic syndrome, which also includes hypertension, insulin resistance, and dyslipidemia, and its increasing prevalence increases the risk of cardiovascular disease and various cancers, leading to substantial public health costs [1]. The association of obesity with obesity-related metabolic comorbidities is explained by chronic low-grade inflammation in the pro-inflammatory adipose tissue macrophages (ATMs) response to the accumulation of apoptotic adipocytes in an expanded adipose tissue (AT) environment [2,3]. Therefore, current anti-obesity strategies require not only weight loss, but also therapeutic control of obesity-induced inflammation.

Berberine (BBR), a stable benzylisoquinoline alkaloid, is a major bioactive component of *Rhizoma Coptidis*, (known as Huang-Lian in Chinese) a traditional herb. It has a wide spectrum of pharmacological properties that are used to treat various diseases, including inflammatory diseases, neurological disorders [4], cardiovascular diseases [5], metabolic diseases, and cancers [6]. Many studies have reported the therapeutic benefits of BBR in the treatment of obesity [7] and its comorbidities, such as diabetes [8] and hyperlipidemia [9,10]. Its anti-obesity effects are related to the amelioration of inflammation [11,12], inhibition of adipogenesis [13], promotion of energy expenditure [14], and regulation of gut microbiota [7]. Previous studies have suggested that BBR exerts immunomodulatory effects on ATMs via TLR4/MyD88/NF-ĸB signaling [11], MAPK signaling [15], and AMPK pathways [12]. Lin et al. [14] found that BBR reduces ATMs and polarizes M2 macrophages, and Ye et al. [16] reported that BBR inhibits M1 macrophage activation. However, the mechanisms underlying chemotaxis in ATMs have not been clearly elucidated. We investigated the anti-obesity effects and underlying mechanisms of BBR with respect to manipulating ATMs-leading inflammation via chemotaxis suppression.

## 2. Materials and Methods

### 2.1. In Silico Study of BBR

#### 2.1.1. Target Genes of BBR

The pharmacological and molecular characteristics of BBR were collected from PubChem (https://pubchem.ncbi.nlm.nih.gov/, accessed on 6 December 2021), SwissADME (https://www.swissadme.ch/, accessed on 6 December 2021) and ADMElab (https://admet.scbdd.com/, accessed on 6 December 2021). We used Lipinski’s rule to examine drug likeness and collected the target genes of BBR using SwissTargetPrediction (http://www.swisstargetprediction.ch/, accessed on 6 December 2021).

#### 2.1.2. Protein-Protein Interaction (PPI) Network Construction

We inspected target genes treating obesity on Genecards, OMIM and DisGeNET using the keyword “obesity.” To list potential obesity-associated genes, we selected genes with relevance scores > 10 from the results of Genecards and added the results from OMIM and DisGeNET. Finally, we uploaded the gene lists to VENNY 2.1 (https://boiinfogp.ncb.csic.es/tools/venny/, accessed on 6 December 2021) to find common genes between BBR and obesity. To construct a PPI network between the overlapping genes, we configured the highest confidence scores in STRING-DB v. 11.5 (https://string-db.org, accessed on 6 December 2021) and selected the top 10 genes with the highest degree scores. The PPI network diagram was illustrated using the Cytoscape software v. 3.9.1 (The Cytoscape Consortium, Boston, MA, USA).

#### 2.1.3. Functional Pathway Analysis

We analyzed the Kyoto Encyclopedia of Genes and Genomes (KEGG) pathway and gene ontology (GO) using the Database for Annotation, Visualization and Integrated Discovery (DAVID) v. 6.8 (http://david.ncifcrf.gov, accessed on 6 December 2021). The dot plots of the GO and KEGG pathway results were presented by R software (R Foundation for Statistical Computing, Vienna, Austria).

#### 2.1.4. Molecular Docking Study

To prepare the molecular docking, we searched a 3D molecular structure of BBR (CID: 2353) from PubChem and used the PyMOL and AutoDock programs to adjust the file format to PDBQT. The structures of proteins were collected from the Protein Data Bank (https://www.rcsb.org, accessed on 6 December 2021): M1 macrophages (PDB ID: 1GD0), M2 macrophages (PDB ID: 1JIZ), tumor necrosis factor-α (TNF-α) (PDB ID:2AZ5), C-C motif chemokine ligand 2 (CCL2) (PDB ID: 1DOK), and C-X-C motif chemokine receptor 4 (CXCR4) (PDB ID: 3ODU). We studied BBR-receptor docking using AutoDock Vina (Center for Computatioinal Structural Biology, La Jolla, CA, USA) and the Biovia Discovery Studio Visualizer (Dassault Systemes, Paris, France).

### 2.2. In Vivo Evaluation of Anti-Obesity Effects and Mechanisms of BBR

#### 2.2.1. Experimental Study Design

Twenty 6-week-old male C57BL/6 mice (Central Lab Animal, Seoul, Korea) were supplied with food and water ad libitum under controlled light, humidity, and temperature. We split the animals into 4 groups; normal chow (NC), control [high fat diet (HFD)], BBR, and metformin (MET) groups. After one-week adaptation, all groups, except NC group, were fed a 60% kcal fat containing HFD (Research Diets D12492; 5.24 kcal/g) to generate an obese model (Appendix A). After 7 weeks, the mice were treated with oral intervention once daily for 10 weeks (100 mg/kg of BBR and 200 mg/kg of metformin). The control group was treated with normal saline using an oral zonde. We purchased BBR from Tokyo Chemical Industry (Tokyo, Japan) and metformin from Sigma-Aldrich (St. Louis, MO, USA). Kyung Hee Medical Animal Research Ethics Committee (KHMC-IACUC) approved all procedures.

#### 2.2.2. Experimental Protocol on Glucose and Lipid Metabolism 

To study the pharmacological effects of BBR on hyperglycemia, an oral glucose tolerance test (OGTT) was performed examining blood glucose levels at 0, 30, 60, 120, and 180 min time-points after oral feeding of 2 g/kg glucose using ACCU-CHECK Performa (Roche, Basel, Switzerland). We measured serum insulin levels using an ultrasensitive mouse insulin ELISA kit (Crystal Chem Inc., Elk Grove Village, IL, USA) and computed the homeostatic model assessment for insulin resistance (HOMA-IR) value as follows: HOMA-IR = fasting blood glucose (mg/dL) × fasting insulin (ng/mL) × 0.071722516669606.

To explore the benefits of BBR on lipid metabolism, an oral fat tolerance test (OFTT) was performed using Accutrend Plus (Roche) and Accutrend (Roche) triglyceride strips. Triglycerides (TG) levels were examined at 0, 120, 240, and 360 min after administration of 2 g/kg olive oil (Sigma, St. Louis, MO, USA). At 16 weeks, serum lipid profiles and safety parameters, including non-esterified fatty acids (NEFA), TG, total, low-density and high-density cholesterol (TC, LDL-C, HDL-C), and phospholipids, aminotransferase (AST), alanine aminotransferase (ALT), and creatinine, were examined from heart blood samples using an ELISA kit (MyBioSource, San Diego, CA, USA) [17].

#### 2.2.3. Quantitative Real-Time Polymerase Chain Reaction (qRT-PCR) 

To extract high quality cellular RNA, we packaged the epididymal fat samples with aluminum foil, kept them at −70 °C, and used Mini RNA Isolation IITM (ZYMO Research, CA, USA). The samples were crushed in 300 μL of ZR RNA buffer, and only the supernatant was collected, centrifuged, and washed twice. After adding RNase-free water, we centrifuged the samples at 100 rpm and kept them at −70 °C [17].

We conducted qRT-PCR on AT samples using a 7900 HT Fast Real-Time PCR System (Applied Biosystems^®^, Waltham, MA, USA) to quantify the expression of inflammatory genes, including TNF-α, F4/80, CCL2, C-C motif ligand 4 (CCL4), C-C motif ligand 5 (CCL5), regulated on activation, normal T-cell expressed, and secreted (RANTES)], and CXCR4. In the process of cDNA synthesis using Advantage RT PCR Kit (Clontech, Palo Alto, CA, USA), we cultured 1 μg of RNA, oligo (dT), and RNase-free H_2_O mixtures at 70 °C for 2 min, kept them at 42 °C for 60 min, and mixed them with MMLV reverse transcriptase, 5× reaction buffer, recombinant RNase inhibitor, and 10 nM dNTP. We finally conducted RT-PCR with dH_2_O, 2× SYBR reaction buffer, and specific primers (Appendix A). We set the cycle threshold (Ct) for relative quantitation based on GAPDH using SDS Software 2.4 (Applied Biosystems^®^) and adjusted the fold-change assuming that the value of the NC group was 1.

#### 2.2.4. Fluorescence-Activated Cell Sorting (FACS) 

Isolated fat tissue samples in 1–2 mm pieces were mixed with phosphate-buffered saline (PBS; Gibco, MA, USA) and 2% bovine serum albumin (BSA; Gibco) and shaken with collagenase (Sigma-Aldrich) and DNase I (Roche). We centrifuged the samples that were filtered through a 100 μm filter, discarded the upper solution and added PBS and 2% fetal bovine serum (FBS; Sigma) to the pellet. Finally, stromal vascular cells remained after centrifugation at 200× *g*. We allowed 106 cells in each sample to react with fluorophore-conjugated antibodies: CD45-APC Cy7 (BioLegend, San Diego, CA, USA), F4/80-APC (BioLegend), CD11c-phycoerythrin (CD11c-PE, Bio-519 Legend, USA), and CD206-FITC (BioLegend). We washed out the samples with 2% FBS/PBS solution and delivered them to FACS tubes. We used a FACS Calibur (BD Biosciences, San Diego, CA, USA) and FlowJo (Tree Star, Inc., Ashland, OR, USA) to estimate the proportion of a total, CD11c(+) and CD206(+) ATMs [17].

#### 2.2.5. Histological Analysis

We fixed the adipose and liver tissue samples in 10% formalin and dehydrated them using 70%, 80%, 95%, and 100% ethanol. Samples on 4 μm-thick gelatin-coated slides were dewaxed, rehydrated, and stained with hematoxylin and eosin (H&E). We captured images using a high-resolution camera-mounted optical microscope (Olympus BX-50, Olympus Optical, Tokyo, Japan), and evaluated the fat area in AT and liver using ImageJ software.

#### 2.2.6. Statistical Analysis

All data are presented as the mean ± standard error of the mean (SEM), and the difference between groups was assessed using GraphPad PRISM 5 (GraphPad Software Inc., San Diego, CA, USA) using one-way analysis of variance (ANOVA) and Tukey’s post hoc test or Mann–Whitney test. Statistical significance was defined as a *p*-value of <0.05. Asterisks (*) represent the statistical significance relative to the NC group and the number of signs (*, #) represent the statistical difference relative to the control group (*, # for *p* < 0.05; **, ## for *p* < 0.01; and ***, ### for *p* < 0.001).

## 3. Results

### 3.1. Network Pharmacological Analysis

#### 3.1.1. Protein-Protein Interaction (PPI)

BBR showed 0.55 of oral bioavailability, −4.809 of Caco-2, and 0 violation for Lipinski’s rule of drug likeness, and inhibited CYP450 1A2, 2D6, and 3A4. We collected 100 target genes of BBR, excluding those with a probability of 0. We also listed 2915 obesity-associated genes from the three databases and figured out 52 overlapping genes between obesity and BBR (Figure 1a). As a result of PPI analysis, the PPI network with highest confidence (0.900) and the 10 highest connected genes are shown in Figure 1b,c.

#### 3.1.2. Functional Enrichment Analysis—KEGG Pathway and GO

We identified KEGG pathways and extracted the top 29 pathways by gene count, including the chemokine signaling pathway, TNF, and PI3K-Akt signaling pathway (Figure 1d). GO analysis revealed 39, 104, and 26 terms in the molecular function (MF), biological process (BP) and cellular component (CC) categories, respectively. We selected significantly enriched terms and extracted 8 MF, 12 BP, and 9 CC terms with involved gene counts (Figure 1e). In the MF category, the activities of the target proteins at the molecular level included protein binding, ATP binding, protein kinase activity, and enzyme binding. In the BP category, the target proteins displayed signal transduction, inflammatory response, oxidation-reduction process, activation of MAPK activity, and innate immune response. In the CC category, the functional locations of the target proteins were described as the cytosol, cytoplasm, plasma membrane, and nucleus.

#### 3.1.3. Molecular Docking Studies

BBR showed high-affinity binding to five receptors (M1, M2 macrophages, TNF-α, CCL2, and CXCR4) (Figure 2a–e). BBR much strongly docked to M2 macrophage by −7.0 kcal/mol, to TNF-α by −8.7 kcal/mol, and to CXCR4 by −7.9 kcal/mol.

### 3.2. Experimental Evaluation of Anti-Obesity Benefits

#### 3.2.1. Change in Weight

Compared with the NC group, the control group showed significant weight gain in the body, epididymal fat pads, and liver. The weights of body, fat pads, and liver were significantly lower in the BBR group than in the control group (body, 44.7 ± 2.74 g vs. 36.92 ± 1.31 g, *p* < 0.05; fat pads, 2.12 ± 0.13 g vs. 1.45 ± 0.01 g, *p* < 0.05; liver, 1.67 ± 0.06 g vs. 1.03 ± 0.05 g, *p* < 0.01), although the food intake in both groups was similar (Figure 3).

#### 3.2.2. Effects on the Size of Fat Area

We found that the size of adipocytes and the fat area in the liver were significantly increased in the control group compared to those in the NC group. The BBR group showed significantly smaller adipocytes compared to the control group (2566 ± 110.3 vs. 3317 ± 221, *p* < 0.001), and reduced fat area in the liver compared to the control group (18 ± 0.76 vs. 46.34 ± 2.12, *p* < 0.001) (Figure 3).

#### 3.2.3. Effects on ATMs

The number of total and CD11c+ ATMs was significantly enlarged in the control group compared to the NC group. Total ATMs were significantly decreased in both BBR and MET groups compared to the control group (61.92 ± 1.44, 63.7 ± 1.31 vs. 82.56 ± 0.99, *p* < 0.001) (Figure 4a–c). While CD11c+ ATMs population displayed no significant difference between HFD-fed groups, the BBR group showed significantly higher values of CD206+ ATMs than the control group (CD206+, 72.85 ± 2.90 vs. 39.3 2.26, *p* < 0.01) (Figure 4c,d).

#### 3.2.4. Effects on Chemokines and Inflammatory Cytokines

We observed that the gene expression of TNF-α, F4/80, CCL2, CCL4, CCL5, and CXCR4 was significantly upregulated in the control group. BBR administration significantly downregulated gene expression of each chemokine and cytokine (Figure 4e–j). The MET group showed a significant reduction in TNF-α expression.

#### 3.2.5. Effects on Hyperglycemia and Hyperlipidemia

To investigate the metabolic benefits of BBR, we performed OGTT and OFTT and estimated HOMA-IR and serum lipids levels. The AUC of the OGTT was significantly higher in the control group than in the NC group (45,975 ± 2380.73 vs. 31,990 ± 1250.53, *p* < 0.001). Glucose levels at 30, 60, and 180 min were significantly lower in the BBR and MET groups than in the control group. In addition, both the BBR and MET groups displayed significantly smaller AUC than the control group (31,686 ± 1544.30 vs. 45,975 ± 2380.73, *p* < 0.001; 28,146 ± 1288.15 vs. 45,975 ± 2380.73, *p* < 0.001) (Figure 5a,b). Fasting blood glucose levels and HOMA-IR in the BBR and MET groups were reduced, compared to those in the HFD group (Figure 5e,f).

In the OFTT, the control group showed a higher AUC than the NC group. However, the BBR group had a significantly smaller AUC than the control group (66,456 ± 2408.07 vs. 144,120 ± 9569.33, *p* < 0.001). TG levels at all time-points were lower in the BBR group than in the control group (Figure 5c,d). In the lipid profile, the BBR group showed significant benefits in NEFA and TG levels, compared to the control group (NEFA, 2157.8 ± 13.11 vs. 2506 ± 32, *p* < 0.001; TG 128 ± 6.68 vs. 173.6 ± 7.53, *p* < 0.05) (Figure 5g,h). TC, LDL-C, and phospholipid levels were also lower in the BBR group than in the control group (Figure 5).

#### 3.2.6. Safety

We observed elevated liver enzymes and serum creatinine levels in the control group compared to those in the NC group. The BBR group showed significantly decreased ALT levels than the control group (24.6 ± 2.90 vs. 52.8 ± 10.71, *p* < 0.05). In addition, the AST and creatinine levels were lower in the BBR group than in the control group (Appendix A). The MET group showed no significant changes in any of the safety parameters, compared to the control group.

## 4. Discussion

We aimed to study the underlying mechanisms of anti-obesity effects of BBR using the network pharmacological method, and to verify the results in vivo using diet-induced obese mice. First, we identified BBR-target protein interactions and key target pathways for the treatment of obesity in silico. Next, we processed an in vivo study to evaluate the metabolic benefits and examine the cellular and molecular changes associated with AT inflammation. Our findings suggested that BBR ameliorates obesity-induced inflammation by regulating macrophage infiltration and polarization in AT, which resulted from the suppression of chemokine signaling pathways, including CCL2 and CXCR4. Our study is the first study to reveal that BBR treats obesity-induced inflammation through chemotaxis modulation, in cellular, molecular, and genetic ways.

In the PPI analysis, we identified the top 10 therapeutic targets of BBR for obesity treatment (in the order of highest degree: SRC, CDC42, RAC1, PIK3CB, JAK2, MAPK14, CDK4, PTPN1, PIK3CG, and ROCK1). SRC (steroid receptor coactivator) enhances pro-inflammatory cytokine production of ATMs in diet-induced obesity including IL-17 [3,18]. CDC42 (cell division cycle 42) and RAC1 (Ras-related C3 botulinum toxin substrate 1) are members of the Rho GTPases family, and the CDC42/RAC pathway specifically activates TLR-induced phagocytosis [19]. CDC42 is pivotal for the translocation of insulin granules [20]. RAC1 is expressed in insulin-sensitive tissues, including AT and skeletal muscle, while activated RAC1 induces oxidative stress in β-cells in obesity [21]. PI3KCB (phosphatidylinositol 4,5-bisphosphate 3-kinase catalytic subunit β) is involved in the signal transduction of insulin and immune response to interleukin 2. It has also been reported that inhibition of PI3KCG (PI3K p110γ) reduces the number of infiltrated M1 macrophages in adipose and liver tissues [22,23]. JAK2 (Janus kinase 2) is predominantly expressed by macrophages and is an important mediator in cytokine-dependent signal transduction, activating SRC-kinase, MAPKs, and PI3K-Akt signaling pathways [24]. MAPK14 (mitogen-activated protein kinase 14), also called p38-α, is crucial for the immune response in various cell types and critically regulates inflammatory cytokines and chemokines in activated macrophages, including TNF-α, IL-6, CCL2, and CCL4 [25]. Deletion of CDK4 (cyclin-dependent kinase 4) is known to increase lipolysis and impair insulin signaling in adipocytes [26]. PTPN1 is known as PTP1B (protein-tyrosine phosphatase 1B) and regulates body fat storage and insulin sensitivity in skeletal muscles [27]. ROCK1 (Rho-associated, coiled-coil-containing protein kinase 1) is a key downstream effector of Rho GTPases that suppresses the recruitment and migration of macrophages and neutrophils during inflammation by binding to PTEN [28]. The functional annotations of the KEGG pathway indicated chemokine signaling and TNF-α and PI3K-Akt signaling pathways. Obese individuals show high levels of TNF-α, which interferes with insulin signaling pathways and lipid metabolism and enhances the expression of chemokines that attract circulating monocytes. TNF-α, a potent pro-inflammatory cytokine, shifts the phenotype of ATMs from M2 to M1, thus worsening adiposity to insulin resistance [29]. Increased free fatty acids levels determined by obesity and TNF-α induce inflammatory cytokines and chemokines, thus establishing a vicious cycle [30]. The PI3K/AKT signaling pathway is associated with glucose homeostasis, lipid metabolism, cell proliferation, and protein synthesis and is damaged by diet-induced obesity, which, in turn, exacerbates insulin resistance. Therefore, the PI3K-AKT pathway has been suggested as an effective and safe target for anti-obesity intervention [31]. Taken together, BBR-targeted therapeutic proteins are related to chemokine signaling and inflammatory cytokine pathways. This was in the same context as the results of KEGG and GO enrichment analysis, and the BP category of GO additionally suggested an innate immune response. We selected the major innate immune cells in obesity-induced inflammation, ATM, ATM-related chemokines and the receptor as the target protein, in the ligand–protein docking simulation.

BBR showed several significant effects on obesity and its comorbidities. First, oral treatment with BBR resulted in the loss of body, fat pads, and liver weights without any change in food intake, and improved fat expansion and liver steatosis. Second, the BBR group showed improved OGTT results and reduced HOMA-IR values. Third, the BBR group had improved OFTT results and reduced TG and NEFA levels. Hu et al. found that BBR significantly improved TG and TC levels, whereas we observed only significant improvements in TG and NEFA levels. As NEFA delivered to the liver are substituted with acyl-CoA, which is used to synthesize TG or cholesterol depending on transcription factors, our results are expected to contribute to improvements in cholesterol levels. Our study also showed a significant NEFA reduction and no significant decrease in TC and LDL-C levels.

Macrophages are critical effector cells in the development of obesity, and their metabolism is a promising therapeutic target [32]. ATMs are the main leukocytes in AT and present as CD45+ F4/80+ [33]. We evaluated the populations of total ATMs and two major subsets of ATMs, and classically activated M1 and alternatively activated M2, which are marked by F4/80+ CD11c+ and F4/80+ CD206+, respectively. In the case of obesity, ATMs transition from the M2 phenotype to the M1 phenotype and an increase in the number of M1 ATMs occur. M1 ATMs form the “crown like” structure around dead adipocytes and produce TNF-α, IL-6, and CCL2, accelerating chronic inflammation and insulin resistance [34]. ATMs in lean individuals express M2 polarization, which is involved in tissue remodeling and secretion of anti-inflammatory cytokines, such as IL-10 and TGF-β [35,36]. Metformin has been reported to suppress ATM infiltration and M1 polarization, which was also shown in our study, with no effect on M2 polarization [37]. However, we found that BBR significantly diminished the ATM population along with the suppression of F4/80 and TNF-α and enhanced M2 ATMs polarization, suggesting the therapeutic potential of BBR in AT inflammation. These findings support the results of previous in vivo and in vitro studies [14,16,38], whereas the effect on the reduction of the M1 phenotype was not significant in our study.

Obese individuals show an increased number of ATMs, which is mainly attributed to macrophage infiltration from trafficking monocytes [2]. Adipocytes secrete pro-inflammatory cytokines that recruit monocytes into AT, which also contributes to a fourfold expansion of ATM and M1 dominant phenotype, worsening obesity and its inflammatory environment [2,35]. Chemokines can be promising therapeutic targets. However, the effects of BBR on chemotaxis remain poorly understood, while a few studies reported the inhibitory effects of BBR on TNF-α and MCP-1 expression [16,39]. Yang et al. [40] reported that BBR reduces CXCR4 expression in diet-induced NASH ApoE mice. Here, we examined the chemotaxis of macrophages by measuring the mRNA levels of TNF-α, F4/80, CCL2, CCL4, CCL5, and CXCR4. The BBR group showed significant suppression of gene expression for all indicators. CCL2 (also known as MCP-1) is an important chemoattractant for monocytes, dendritic cells, natural killer cells, and T cells in various inflammatory diseases. In HFD-fed obesity, CCL2 expression increased, followed by large recruitment of ATMs and increased pro-inflammatory cytokines; furthermore, the phenotype switch of ATM toward M1 depended on the expression of CCL2 [33]. CCL4, also known as macrophage inflammatory protein-1β (MIP-1 β), is a macrophage attractant that induces metabolic inflammation, and is upregulated by palmitate via the NF-κB/MAPK/PI3K signaling pathway [41]. CCL5, regulated on activation, normal T-cell expressed, and secreted (RANTES), increases the number of ATMs and inhibits the switch of ATM toward an M2 phenotype. In previous studies, CCL5 was found to have higher AT levels in obese individuals than in lean individuals [42]. CXCR4 is a receptor specific to CXCL12, which controls leukocyte recruitment into the AT and the functional response of adipocytes. The CXCL12-CXCR4 axis supports the thermogenic response in brown adipose tissue, while inducing M1 ATM accumulation and worsening insulin resistance in white adipose tissue [43]. We first observed significant inhibitory effects on chemotaxis in HFD-induced obesity, which was consistent with in silico results of BBR-related pathways. Taken together, these results suggested that BBR suppresses TNF-α, CCL2, CCL4, and CCL5 expression via CXCR4/NF-ĸB signaling. Lee et al. [9] explained the effects of BBR on body weight and TG levels by regulating mitochondrial biosynthesis via the AMPK pathway. However, our results suggest that the metabolic effects of BBR result from the anti-inflammatory effects at the macrophage level by chemotaxis inhibition. Our new findings would support the clinical benefits of oral administration of BBR to treat obesity and its various metabolic comorbidities.

## Figures and Tables

**Figure 1 biomedicines-10-01739-f001:**
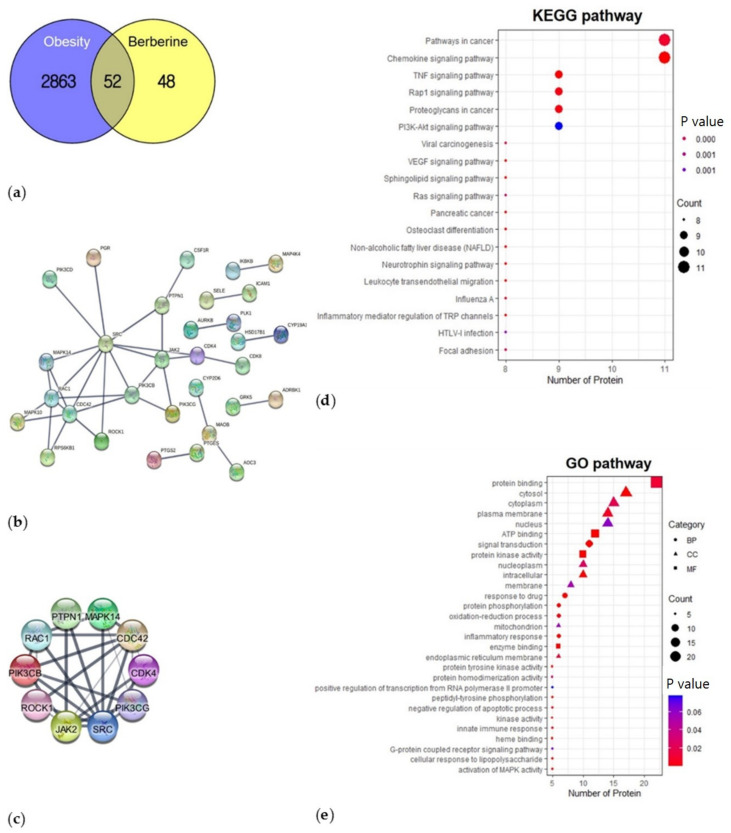
Therapeutic targets of BBR to treat obesity and the functional annotation analysis. (**a**) The 52 overlapping genes between BBR and obesity; (**b**) PPI network; (**c**) Top 10 genes with highest connectivity; (**d**) KEGG pathways analysis; (**e**) GO enrichment analysis. BBR: berberine, PPI: protein-protein interaction, KEGG: Kyoto Encyclopedia of Genes and Genomes, GO: gene ontology.

**Figure 2 biomedicines-10-01739-f002:**
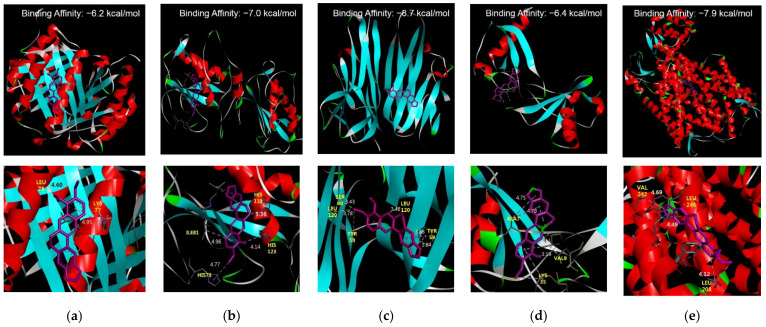
Molecular docking simulation. (**a**) BBR and M1 macrophage; (**b**) BBR and M2 macrophage; (**c**) BBR and TNF; (**d**) BBR and CCL2; (**e**) BBR and CXCR4. M1 macrophage (PDB ID: 1GD0), M2 macrophage (PDB ID: 1JIZ), TNF-α (PDB ID: 2AZ5), CCL2 (PDB ID: 1DOK) and CXCR4 (PDB ID: 3ODU). Binding energy values are written on the top of each subfigure. BBR: berberine, TNF: tumor necrosis factor, CCL2: C-C motif chemokine ligand 2, CXCR4, C-X-C motif chemokine receptor 4.

**Figure 3 biomedicines-10-01739-f003:**
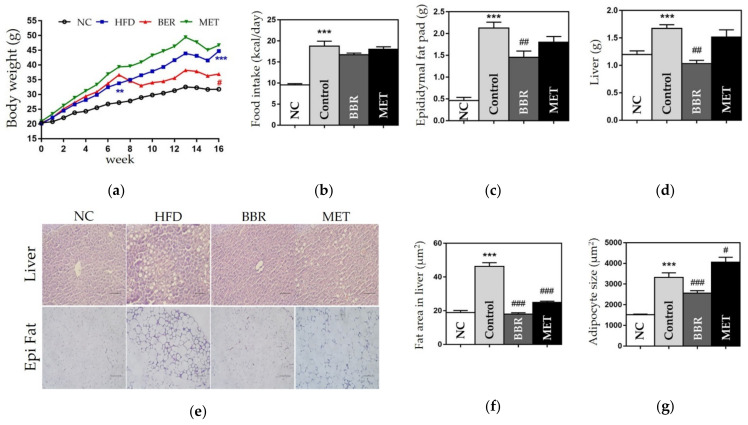
Changes in body weight, organ weight, and epididymal fat pads weight, and the fat area in the liver. (**a**) Body weight; (**b**) Food intake; (**c**) Epididymal fat pads weight; (**d**) Liver weight; (**e**) Histological results of the liver and epididymal fat pads; (**f**) Fat area in the liver; and (**g**) The size of adipocyte. In histological images stained by hematoxylin and eosin (H&E), the scale bar indicates 5 μm in the liver and 100 μm in the epididymal fat. Data are presented as mean ± standard error of the mean (SEM). *** *p* < 0.001 vs. the NC group and # *p* < 0.05, ## *p* < 0.01 and ### *p* < 0.001 vs. the control group. NC: normal chow, HFD: high fat diet, BBR: berberine, MET: metformin.

**Figure 4 biomedicines-10-01739-f004:**
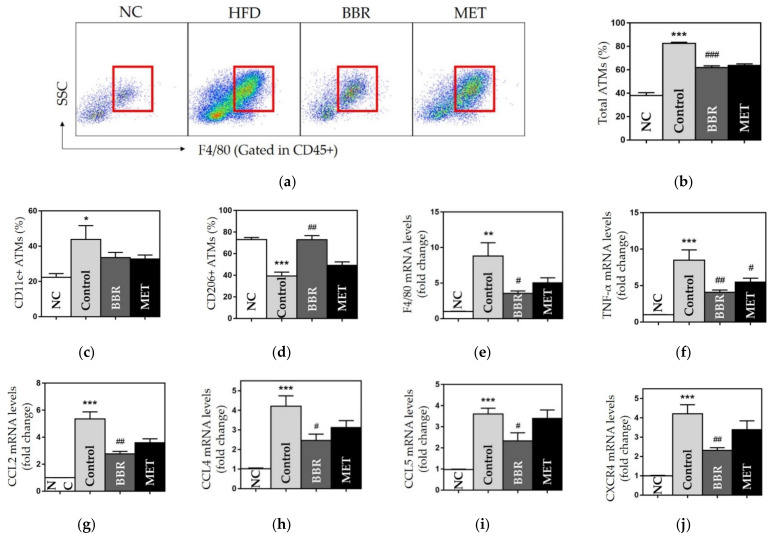
The population and polarization of ATMs and the gene expressions of chemokines. (**a**) Flow cytometry results and gating strategy of CD45+ & F4/80+ ATMs (red box); the populations of (**b**) CD45+ & F4/80+ ATMs; (**c**) CD11c+ ATMs; (**d**) CD206+ ATMs; and the gene expressions of (**e**) F4/80; (**f**) TNF-α; (**g**) CCL2; (**h**) CCL4; (**i**) CCL5; and (**j**) CXCR4. Data are presented as mean ± standard error of the mean (SEM). * *p* < 0.05, ** *p* < 0.01 and *** *p* < 0.001 vs. the NC group and # *p* < 0.05, ## *p* < 0.01 and ### *p* < 0.001 vs. the control group. ATMs: adipose tissue macrophages, NC: normal chow, HFD: high fat diet, BBR: berberine, MET: metformin, TNF-α: tumor necrosis factor α, CCL2: C-C motif chemokine ligand 2, CCL4: C-C motif chemokine ligand 4, CCL5: C-C motif chemokine ligand 5, CXCR4, C-X-C motif chemokine receptor 4.

**Figure 5 biomedicines-10-01739-f005:**
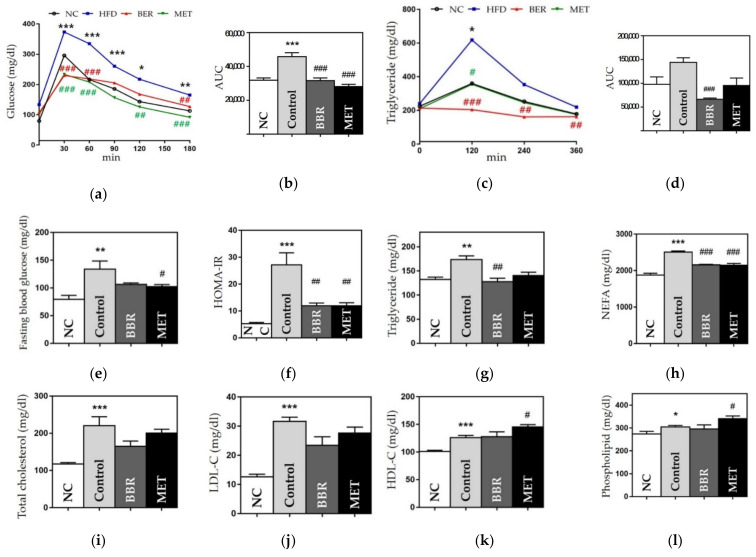
Glucose- and lipid metabolism-related outcomes. (**a**) OGTT results, (**b**) AUC of OGTT, (**c**) OFTT results, (**d**) AUC of OFTT, (**e**) fasting blood glucose, (**f**) HOMA-IR (**g**) triglycerides (**h**) NEFA, (**i**) total cholesterol level, (**j**) LDL cholesterol level, (**k**) HDL cholesterol level, and (**l**) phospholipid level. Data are presented as mean ± standard error of the mean (SEM). * *p* < 0.05, ** *p* < 0.01 and *** *p* < 0.001 vs. the NC group and # *p* < 0.05, ## *p* < 0.01 and ### *p* < 0.001 vs. the control group. AUC: area under curve, OGTT: oral glucose tolerance test, OFTT: oral fat tolerance test, HOMA-IR, homeostatic model assessment for insulin resistance, NEFA: non-esterified fatty acid, LDL: low-density lipoprotein, HDL: high-density lipoprotein NC: normal chow, HFD: high fat diet, BBR: berberine, MET: metformin.

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
