# Peer review of "Cellular and Molecular Mechanisms and Effects of Berberine on Obesity-Induced Inflammation"

_biomedicines, 2022, doi:10.3390/biomedicines10071739_

Round 1

Reviewer 1 Report

The manuscript entitled "Cellular and Molecular Mechanisms and Effects of Berberine 2 on Obesity-Induced Inflammation" incorporates new findings that allow a better understanding of the inflammatory process in obesity, as well as its possible treatments.

However, some points should be clarified:

 What is the composition of the hypercaloric diet? quantities, type of fat, etc. That was provided to the HD group

 Lines 104-106

What information can the short-term oral fat intake tolerance test provide? Why was a source of saturated fat not selected? Considering that metabolic imbalance due to excess fat intake is a process of months or years, to develop obesity and consequently low-grade inflammation.

Lines 160,161

3.1.1. Protein-Protein Interaction (PPI) 160

BBR showed 36.86% of oral bioavailability, 1.24 of Caco-2, and 0.78 of drug-likeness, 161 and inhibited CYP450 1A2, 2D6, and 3A4

It would be appropriate to incorporate pharmacokinetic data for berberine.

 The aforementioned, with what methodology was it carried out? From what experimental or theoretical data are these bioavailabilities and/or inhibitions obtained?

According to Fig. 1b and 1c (Protein interaction network obtained)

Why were these not included in the mRNA expression study?

In Fig. 1a, what do those found in pairs or interaction of 3 proteins mean? Why are they not inside the main network?

In other words, in paragraphs 173-183, the selection of proteins according to the signaling pathway is mentioned, but it is not explained in each category. What were the selection criteria for the target proteins?

Why M1, M2, TNFα, CCL2, CCL4, CCL5, and CXCR4 from the 8MF, 12BP, and 9CC?

The selected target proteins, whose function and importance in the process of lipolysis-lipogenesis-inflammation are fully explained in the discussion, however, it is not clear how they were selected.

How do you explain that metformin increases body weight when various studies have shown that it effectively reduces body weight? Fig. 3a

Author Response

Answers to Reviewer Comments

Manuscript No.: biomedicines-1803739

Authors: Ji-Won Noh et al.

Title: “Cellular and molecular mechanisms and effects of berberine on obesity-induced inflammation.”

Thank you very much for considering our manuscript for publication. Your suggestions were very helpful to us, and we have incorporated those points into our revised manuscript.

The changes made to the manuscript are as follows:

The manuscript entitled "Cellular and Molecular Mechanisms and Effects of Berberine on Obesity-Induced Inflammation" incorporates new findings that allow a better understanding of the inflammatory process in obesity, as well as its possible treatments. However, some points should be clarified:

  1. What is the composition of the hypercaloric diet? quantities, type of fat, etc. That was provided to the HD group
  • The HFD provided to the control group was D12492 diet (Research Diets, USA) consisting of a fat source of soybean oil and particularly lard. We attached the supplementary table (Table S1) about the detail composition of high-fat diets and edited the manuscript in line 92.
  1. Lines 104-106. What information can the short-term oral fat intake tolerance test provide? Why was a source of saturated fat not selected? Considering that metabolic imbalance due to excess fat intake is a process of months or years, to develop obesity and consequently low-grade inflammation.
  • OFTT is a reliable method of assessing postprandial lipidemia because it represents the body’ capacity to normalize hyperlipidemia after lipid intake. The response is exaggerated and delayed in subjects with metabolic syndrome causing hypertriglyceridemia.
  • Previous in vivo studies with mice and rats have shown that short-term as 6 weeks of high fat diet feeding can make a condition of metabolic syndrome such as hyperglycemia, hyperlipidemia, and increased-level of inflammatory cytokines. As a source of fat study, HFD is lard, Buettner et al. defined that high fat sourced by lard showed the most pronounced obesity and insulin resistance.

Related References

  • G. Rodrigues, C. V. A. Lutterback-de-Carvalho, D. F. Motta et al., “Oral fat tolerance test in male and female C57BL/6 mice.” Bio Protoc, vol. 11, e4062, 2021.
  • Crinigan, M. Calhoun and K. L. Sweazea. “Short-term high fat intake does not significantly alter markers of renal function of inflammation in young male Sprague-dawley rats.” J Nutr Metab, ID: 157520, 2015. DOI: 10.1155/2015/157520.
  • Buettner, K. B. Parhofer, M. Woenckhaus et al., “Defining high-fat-diet rat models: metabolic and molecular effects of different fat types” J Mol Endocinol, vol. 36, pp. 485-501, 2006.
  1. Lines 160,161 3.1.1. Protein-Protein Interaction (PPI). BBR showed 36.86% of oral bioavailability, 1.24 of Caco-2, and 0.78 of drug-likeness, 161 and inhibited CYP450 1A2, 2D6, and 3A4.

It would be appropriate to incorporate pharmacokinetic data for berberine. The aforementioned, with what methodology was it carried out? From what experimental or theoretical data are these bioavailabilities and/or inhibitions obtained?

  • We obtained the pharmacokinetic data from the database of websites; SwissADME and ADMElab (https://admet.scbdd.com). These websites gather the most relevant computational methods of ADME and pharmacokinetics and predict ADME parameters, pharmacokinetic properties, and drug-likeness (https://admet.scbdd.com/home/modeling process/# Toc469587649). In detail, the human colon adenocarcinoma cell lines (Caco-2), as an alternative approach for the human intestinal epithelium, has been commonly used to estimate in vivo drug permeability. In this study, the dataset of Caco-2 permeability was collected from a QSAR study group that carried out 1182 compounds.

In the case of drug-likeness, we used Lipinski's rule of five, which is a rule of thumb that describes the drugability of a determinate molecule. This rule consists of No more than 5 hydrogen bond donors. No more than 10 hydrogen bond acceptors, Molecular mass less than 500 Da, and Partition coefficient not greater than 5. The violation of 2 or more of these conditions predicts a molecule as a non-orally available drug. It helps to determine the chemical and physical properties, and oral bioavailability of active chemicals. 

The results of CYP450 1A2, 3A4, 2C19, 2C9, and 2D6 inhibition were obtained from the PubChem BioAssay database (AID: 1851), which was quantitatively screened using in vitro bioluminescent assay. 

  • We edited section 2.1.1. and 3.1.1. in the manuscript as explained above.

Related References

  • N. Wang, et al., “ADME Properties Evaluation in Drug Discovery: Prediction of Caco-2 Cell Permeability Using a Combination of NSGA-II and Boosting.” Journal of Chemical Information & Modeling, vol. 56, no. 4, 2016.
  • Veith, et al., “Comprehensive characterization of cytochrome P450 isozyme selectivity across chemical libraries.” Nature Biotechnology, vol. 27, no. 11, pp.1050-1055, 2009.
  1. According to Fig. 1b and 1c (Protein interaction network obtained), why were these not included in the mRNA expression study? In Fig. 1a, what do those found in pairs or interaction of 3 proteins mean? Why are they not inside the main network?
  • About Fig. 1a, among the disease of drug-related genes studies so far, we confirmed the common genes between obesity-related genes and those known to be affected by berberine. The common genes are berberine’s target proteins in obese patients. Also, they were the background of PPI network analysis.
  1. In other words, in paragraphs 173-183, the selection of proteins according to the signaling pathway is mentioned, but it is not explained in each category. What were the selection criteria for the target proteins? Why M1, M2, TNFα, CCL2, CCL4, CCL5, and CXCR4 from the 8MF, 12BP, and 9CC? The selected target proteins, whose function and importance in the process of lipolysis-lipogenesis-inflammation are fully explained in the discussion, however, it is not clear how they were selected.
  • By KEGG analysis, we have identified berberine’s anti-obesity effect is strongly associated with ‘inflammation’. According to the results of GO analysis, most of the related pathways were associated with energy expenditure and inflammatory cytokine signaling, which were already investigated by previous mechanism studies. To select the target proteins for molecular docking in an uninspected field, we examined BP (biological process) category of GO analysis and decided to focus on ATM as a major innate immune cell, ATM-related chemokines and cytokines, and the receptors.

We added two sentences to clarify the process of docking protein selection in lines 315-319 of the discussion part.

  1. How do you explain that metformin increases body weight when various studies have shown that it effectively reduces body weight? Fig. 3a
  • At the end of our study, the difference between MET and the control groups was positive but not significant. However, considering the slope of the graph of the MET group in Fig. 3a got reduced after metformin intervention, and that of the HFD group was consistent throughout the study period, metformin seemed to limit the weight gain.
  • Golay (2008) reviewed that the effects of metformin on body weights are variable in the context of not increasing weight. McPherson (2020)’s study showed that 6 weeks of metformin administration to a 16-week-diet-induced obese mouse model didn’t reduce body weight (HFD group 40.2 g vs. HFD+Metformin group 40.6 g).

Related reference

  • A Golay, “Metformin and body weight”, Int J Obesity, vol. 32, pp. 61-72, 2008.
  • O. McPherson and M. Lane, “Metformin treatment of high-fat-diet-fed obese male mice resotres sperm function and fetal growth, without requiring weight loss.”, Asian J Androl, vol.22, no.6, pp. 560-568, 2020.

We thank you again for your insightful comments on our paper.

                                                       Sincerely yours,

 Byung-Cheol Lee, M.D.& Ph.D.

Reviewer 2 Report

The present manuscript aims at proving that Berberine (Rhizoma Coptidis) decreases Obesity-Induced Inflammation by chemotaxis inhibition. Introduction  presents the background and creates a good hypothesis for the research. I would have liked to see more insight of the recent progress made in the field of obesity.  Please add one more paragraph on the originality and added value of the study while emphasizing more on its practical applications.

Materials and methods are well presented and executed, the study design is appropriate. Discussions are well conducted and they point out the originality of the research. I recommend publication after minor corrections.

Author Response

Answers to Reviewer Comments

Manuscript No.: biomedicines-1803739

Authors: Ji-Won Noh et al.

Title: “Cellular and molecular mechanisms and effects of berberine on obesity-induced inflammation.”

Thank you very much for considering our manuscript for publication. Your suggestions were very helpful to us, and we have incorporated those points into our revised manuscript.

The changes made to the manuscript are as follows:

The present manuscript aims at proving that Berberine (Rhizoma Coptidis) decreases Obesity-Induced Inflammation by chemotaxis inhibition. Introduction presents the background and creates a good hypothesis for the research. I would have liked to see more insight of the recent progress made in the field of obesity.  Please add one more paragraph on the originality and added value of the study while emphasizing more on its practical applications.

 Materials and methods are well presented and executed, the study design is appropriate. Discussions are well conducted and they point out the originality of the research. I recommend publication after minor corrections

  • As you mentioned, we strengthened the less emphasized originality and practical applications by editing the representing sentences in Line 277, 378-380.

We thank you again for your insightful comments on our paper.

                                                       Sincerely yours,

 Byung-Cheol Lee, M.D.& Ph.D.

Round 2

Reviewer 1 Report

The manuscript can be published in its current version. All questions have been properly addressed, improving the requested paragraphs.